# Epidemiology of giardiasis and assemblages A and B and effects on diarrhea and growth trajectories during the first 8 years of life: Analysis of a birth cohort in a rural district in tropical Ecuador

Tannya Sandoval-Ramírez[1,2☯], Victor Seco-Hidalgo[1,3☯], Evelyn Calderon-Espinosa[1☯], Diana Garcia-Ramon[1], Andrea Lopez[1], Manuel Calvopiña[4], Irene Guadalupe[5], Martha Chico[5], Rojelio Mejia[6], Irina Chis Ster[3], Philip J. Cooper [1,3,5]*

1 School of Medicine, Universidad Internacional del Ecuador, Quito, Ecuador, 2 Grupo de Investigación en Sanidad Animal y Humana GISAH, Departamento de Ciencias de la Vida y la Agricultura, Universidad de las Fuerzas Armadas ESPE, Quito, Ecuador, 3 Institute of Infection and Immunity, St George's University of London, London, United Kingdom, 4 One Health Research Group, Faculty of Medicine, Universidad de las Américas, Quito, Ecuador, 5 Fundación Ecuatoriana Para la Investigación en Salud, Quito, Ecuador, 6 National School of Tropical Medicine, Baylor College of Medicine, Houston, Texas, United States of America

☯ These authors contributed equally to this work.
* pcooper@sgul.ac.uk

## Abstract

### Background

There are limited longitudinal data on the acquisition of *Giardia lamblia* infections in childhood using molecular assays to detect and type assemblages, and measure effects of infections on diarrhea risk and childhood growth.

### Methods

We analysed stool samples from a surveillance sample within a birth cohort in a rural district in tropical Ecuador. The cohort was followed to 8 years of age for the presence of *G. lamblia* in stools by quantitative PCR and A and B assemblages by Taqman assay or Sanger sequencing. We explored risk factors associated with infection using generalized estimating equations applied to longitudinal binary outcomes, and longitudinal panel data analysis to estimate effects of infection on diarrhea and growth trajectories.

### Results

2,812 stool samples collected between 1 month and 8 years of age from 498 children were analyzed and showed high rates of infection: 79.7% were infected at least once with peak prevalence (53.9%) at 5 years. Assemblage B was accounted for 56.8% of genotyped infections. Risk factors for infection included male sex (P = 0.001), daycare attendance (P<0.001), having a household latrine (P = 0.04), childhood (P<0.001) and maternal soil-

**Data Availability Statement:** All relevant data are within the paper and its Supporting Information files.

**Funding:** This work was supported by Wellcome Trust (grants 074679/Z/04/Z and 088862/Z/09/Z to PJC) and by the University International del Ecuador (grant EDM-INV-02-19 to PJC, VS-H, and DG-R) The funders had no role in study design, data collection and analysis, decision to publish, or preparation of the manuscript.

**Competing interests:** The authors have declared that no competing interests exist.

transmitted helminth (P = 0.029) infections, and exposures to donkeys (age interaction P = 0.034). *G. lamblia* was associated with increased risk of diarrhea (per episode, RR 1.03, 95% CI 1.01–1.06, P = 0.011) during the first 3 years of life and a transient impairment of weight (age interaction P = 0.017) and height-for-age (age interaction P = 0.025) trajectories between 1 and 4 years of age. There was no increased risk of either assemblage being associated with outcomes.

## Conclusion

Our data show a relatively high edemicity of *G. lamblia* transmission during childhood in coastal Ecuador, and evidence that infection is associated with a transiently increased risk of diarrhea during the first 3 years of life and impairment of weight and height between 1 and 4 years.

## Author summary

*Giardia lamblia* is an intestinal protozoal pathogen that is estimated to infect annually over 200 million people worldwide. The infection typically causes a self-limiting diarrheal illness, but which occasionally may be severe and persistent. The infection is most common in children living in poor regions of the tropics in conditions of inadequate access to clean water and sanitation. In endemic settings, associations of infections with diarrhea in children are variable and there is evidence that subclinical infections may cause growth stunting. Here, we analyzed stool samples collected for a surveillance cohort of children followed up from birth to 8 years of age in a rural district of tropical coastal Ecuador. Stool samples were analyzed for the presence of *G. lamblia* infections using a highly sensitive molecular test (quantitative PCR) and positive samples were genotyped for *G. lamblia* assemblages. *G. lamblia* was detected in almost 80% of children during follow-up and about half the cohort was infected at 5 years of age. Assemblages A and B were detected with B being the most frequent assemblage. Risk factors for infection included male sex and factors associated with increased early contacts with other children (i.e., daycare attendance), poor sanitation (having a household latrine and infections with other enteric parasites), and exposures to donkeys as a potential source of zoonotic infections. There was evidence that early *G. lamblia* infections were associated with effects of increased diarrheal risk up to 3 years and reduced linear and ponderal growth between 1 and 4 years of age. We did not see any relative effect of assemblage (i.e., B vs. A) on disease. Our data, from a longitudinal birth cohort showed a relatively high prevalence of infection in this rural tropical setting that was associated with the possible onset of adaptive immunity against (diarrheal) disease and transient effects on growth trajectories.

## Introduction

Giardiasis, caused by the enteric protozoal parasite *Giardia lamblia* (also known as *Giardia duodenalis* and *Giardia intestinalis*), is a neglected infectious disease with a worldwide distribution [1]. More than 200 million people worldwide are estimated to be infected annually [2]. The parasite forms a genetic complex with eight assemblages (A to H) that differ in host specificity with assemblages A and B being pathogenic in humans [3–5]. *G. lamblia* infections are

most prevalent among children living in the warm tropics in insanitary conditions [5,6]. Infection is acquired through fecal-oral transmission, primarily through contaminated water and food or through direct contact with other infected humans [7,8]. Infections with *G. lamblia* may be asymptomatic [9–12] but frequently cause a self-limiting diarrheal illness in children that may be prolonged, and be accompanied by malabsorption and growth impairment [13].

*G. lamblia* infections are first acquired during infancy in highly endemic regions [14] and rates of asymptomatic or sub-clinical infection may exceed 50% during childhood in Latin American settings [15,16]. Although *G. lamblia* infections cause acute diarrhea in non-endemic settings such as during waterborne outbreaks [17], the relationship with diarrhea in endemic settings is less clear [18]. Giardiasis is considered to cause stunting and/or failure-to-thrive in children living in endemic regions [19,20], although not all studies have been consistent in showing this effect [19]. Most studies looking at effects of giardiasis on diarrhea and/or growth have been cross-sectional with relatively few longitudinal analyses from birth [2,21–23]. A longitudinal approach from birth would allow consideration of the temporal sequence between infections first acquired in early infancy and potentially time-dependent effects on diarrhea and growth impairment.

In the present analysis, we studied the molecular epidemiology of *G. lamblia* and the impact of childhood infections on diarrhea risk and growth trajectories during the first 8 years of life, using a sample from a birth cohort recruited in a tropical region in coastal Ecuador. Stool samples were analyzed for the presence of *G. lamblia* infections by qPCR and genotyped for assemblages A and B. We explored individual, parental, and household factors as determinants of *G. lamblia* infection and the relative risk of infections with assemblages A versus B, as well as the longitudinal effects of *G. lamblia* infections on risk of diarrhea and childhood growth.

## Methods

### Ethics statement

The study protocol was approved by the ethics committees of the Hospital Pedro Vicente Maldonado (2005) and Universidad San Francisco de Quito (2010). Informed written consent was obtained from the child's mother for collection of stool samples and data. Anti-protozoal treatments (metronidazole or tinidazole) were offered to symptomatic children with *E. histolytica* or *G. lamblia* trophozoites in fresh stool samples. Individuals with positive stools for soil-transmitted helminth infections were treated with a single dose of albendazole if aged 2 years or greater and with pyrantel pamoate if aged <2 years, according to Ecuadorian Ministry of Public Health recommendations [24].

### Study area and population

Detailed methodology of the study objectives, design, follow-up and sample and data collection for the ECUAVIDA cohort study is provided elsewhere [25].Briefly, the ECUAVIDA cohort was a population-based birth cohort of 2,404 newborns whose families lived in the rural district of Quinindé, Esmeraldas Province, and were recruited around the time of birth at the Hospital Padre Alberto Buffoni (HPAB) in the town of Quinindé between November 2005 and December 2009. This population-based cohort was designed to study the effects of early life infections on the development of allergy and allergic diseases in childhood. The present analysis focused on a subset of all 504 children enrolled between October 2008 and December 2009. The district of Quinindé is largely agricultural where the main economic activities relate to the cultivation of African palm oil and cocoa. The climate is humid tropical with temperatures generally ranging 23–32°C with yearly rainfall of around 2000–3000 mm. Inclusion

criteria were being a healthy baby, collection of a maternal stool sample, and planned family residence in the district for at least 3 years.

## Study design and sample collection

Children were followed-up from birth to 8 years of age with data and stool samples collected at 1, 3, 7, 13, 18, 24, 30 months, and 3, 5, and 8 years of age. Follow-ups were done either by scheduled visits to a dedicated clinic at HPAB or by home visits. In addition, surveillance for diarrhea was done between birth and 3 years of age through clinic and home visits for evaluation of acute diarrheal illnesses with sampling of symptomatic children. Diarrhea was defined as the passage of three or more liquid or semi-liquid stools in a 24-hour period. At the initial home visit, a questionnaire was administered to the child's mother by a trained member of the study team to collect data on potential risk factors [25]. Maternal questionnaires were repeated at 7 and 13 months and 2, 3, 5, and 8 years of age.

## Anthropometric measurements

Anthropometric measurements were done as described [26]. Briefly, first measurements of weight and height were done between birth and 2 weeks of age and then repeated periodically during clinic and home visits at 7, 13, 24, 36, 60, and 96 months. At each observation time, length/height (cm) and weight (kg) were measured in duplicate by trained members of the research team. Children were weighed without clothes or with light underwear on portable electronic balances (Seca, Germany) accurate to within 100 grams. Length/height measurements were done using wooden infantometers/stadiometers to within 0.1 cm. Z scores for weight-for-age (WAZ), height-for-age (HAZ) and body mass index-for-age (BAZ) at each observation time were calculated using WHO growth standards [27].

## Stool examinations

Stool samples were examined using four microscopic techniques to detect and/or quantify STH eggs and larvae including direct saline wet mounts, formol-ether concentration, modified Kato-Katz, and carbon coproculture [28]. All stool samples were examined using all 4 microscopic methods where stool quantity was adequate. A positive sample for STH was defined by the presence of at least one egg or larva from any of the above detection methods. An aliquot of stool was preserved in 90% ethanol at -30˚C for molecular analyses.

## Molecular analyses for *Giardia lamblia*

Stool samples (50 mg of stool stored in 90% ethanol at -30˚C) were processed using FastDNA SPIN Kit for Soil (MP Biomedicals, Santa Ana, California, USA) [29]. Stool DNA was analysed by quantitative polymerase chain reaction (qPCR) to detect *G. lamblia* [29]. Primers and probes used are shown in S1 Table. All reactions were performed in a total volume of 7 μL containing 3.5 μL of TaqMan fast mix (Applied Biosystems), 1 μL of template DNA, and 0.007 μL of primers (final concentration of 0.9 μM) and 0.0175 μL of FAM-labelled minor groove binder probe (final concentration of 0.25 μM), and 2.469 μL nuclease-free water. Positive *G. lamblia* samples were genotyped either using a novel SNP TaqMan Assay or by 18S rRNA sequencing. All extractions included 1 μl of internal amplification control (IAC) plasmid at a concentration of 100 pg/μl, containing a unique 198-bp sequence and detected by qPCR using PCR primers and probe sequences as described [30].

## 18S rRNA SNP Genotyping Assay

We designed a SNP Taqman assay to identify *G. lamblia* assemblages A and/or B, which are the most common assemblages present in human fecal samples including in Ecuador [5]. Applying the criteria recommended by Applied Biosystems for SNP Taqman assays, we defined the region of interest and identified a consensus sequence for assemblages A and B through multiple alignment using the CLUSTAL Omega algorithm [31,32] for the *G. lamblia* reference sequences (GenBank accession numbers): assemblage A (AF199446, LC437354.1 and LC341259.1) and assemblage B (AF199447, LC341260.1 and MK990739.1). The consensus sequence and the single nucleotide polymorphisms selected to distinguish A from B alleles is shown in S1 Fig. All SNP genotyping reactions were performed in a total volume of 10 µL with 5 µL of TaqMan Master Mix (Applied Biosystems), 0.5 µL of 20x custom TaqMan SNP Geno-typing Assay solution (Applied Biosystems) that includes primers and probes (S1 Table), 1 µL of template DNA, and 3.5 µL of nuclease-free water. Samples were run on an ABI 7500 Fast for 10 mins at 95°C, followed by 40 cycles of 15 secs at 95°C and 1 min at 60°C, and finally at 72°C for 5 mins. Samples with low levels of *G. lamblia* DNA (defined as <28.02 fg/µl equivalent to a Ct value of 33 on qPCR) were genotyped by 18S rRNA sequencing.

## 18S rRNA sequencing and sequence analysis

PCR reactions were developed using Platinum SuperFi PCR Master Mix Kit (Invitrogen) in a total volume of 15 µL containing 7.5 µL of 2x Platinum SuperFi PCR Master Mix, 0.75 µL of BSA (final concentration of 0.1 µg/µL), 0.6 µL of 25mM $MgCl_2$, 3 µL of 5x SuperFi GC Enhancer, 1.2 µL of template DNA, 0.38 µL of primers (final concentration of 0.5 µM), and 1.2 µL of nuclease-free water. The primers used were RH11 and RH4 as described (S1 Table) [33]. Sample amplification was optimized in this study by a touchdown PCR by heating the reactions to 96°C for 2 min followed by 8 cycles of 96°C for 20 sec, 71–64°C ΔT:1°C for 20 sec, and 72°C for 30 sec, followed by 30 cycles of 96°C for 20 sec, 63°C for 20 sec and 72°C for 30 sec, and 1 cycle of 72°C for 7 min using an ABI 7500 Fast. Templates were purified and sequenced by MACROGEN (Seoul, South Korea). Sequences were analysed with Snapgene viewer. Identification of *G. lamblia* assemblages was done using BLASTN (**https://blast.ncbi. nlm.nih.gov/Blast.cgi**) with GenBank reference sequences: assemblage A (AF199446), assemblage B (AF199447), assemblage C (AF199449), assemblage D (AF199443), assemblage E (AF199448), assemblage F (AF199444), assemblage G (AF199450). Mixed infections were determined by sequencing alignment using Clustal W in the program Molecular Evolutionary Genetics Analysis- MEGA 7 (MEGA Software). Mixed infections were identified by the presence of double peaks on electropherograms and the presence of ambiguous bases in SNP positions following alignment with reference sequences. Samples with mixed infections with assemblages A and B were confirmed using the 18S rRNA SNP Genotyping Assay.

## Statistical analytic strategy

Sample size for analyses within this surveillance sub-sample of the birth cohort was determined by logistic and cost considerations. We generated two longitudinal statistical outcomes for *G. lamblia* infection (binary for presence/absence of *G. lamblia* for each child during childhood and 4-categorical for specific genotype when *G. lamblia* was present [i.e absence of infection, A, B, and mixed A and B]). Longitudinal analyses were done as described previously [34]. Potential risk factors considered included individual (for participant child—sex, age, birth order, duration of breast feeding and exclusive breast feeding, day care during the first 3 years of life, time-varying acquisition of soil-transmitted helminths [any STH and infections with *Ascaris lumbricoides* or *Trichuris trichiura*], receipt of anthelmintic treatments [time-varying]),

parental (age, ethnicity, educational status, and presence of STH infections around the time of the child's birth), and household (area of residence, socioeconomic status, overcrowding, monthly income, construction materials, material goods [fridge, TV radio, and HiFi], type of bathroom [time-varying], agricultural activities, and presence of peri-domiciliary animals [time-varying] including dogs, cats, pigs, chickens, cows, and equines [horses, donkey and/or mules] and presence of STH infections in household members around the time of the child's birth). Socioeconomic variables were combined to create a socio-economic status (SES) index using principal components analysis for categorical data as described [35]. We used generalized estimation equations models (GEE) to fit population-averaged models [36] to understand the effects of age, childhood, parental and household characteristics, on the age-dependent risk of acquiring *G. lamblia* in childhood. Binary random effects models were also considered [37,38]—a lay description of the two approaches is provided elsewhere [38]. The GEE approach provides population-average while random effects analysis gives subject-specific estimates. The latter was particularly useful for time-varying exposures and can indicate the effect of change in an explanatory factor on risk *of G. lamblia* infection. Because of the similarity of the estimates obtained and the fact that our questions were addressed at the population level, we have presented population average estimates only. Associations and their uncertainties were measured by odds ratios (OR) and their corresponding 95% confidence intervals (CIs). ORs derived from these longitudinal models estimated associations between potential explanatory variables and the age-dependent risk of *G. lamblia*, and their interpretation is similar to that of a cross-sectional OR. Age-adjusted models (for age in polynomial forms to the power of 5) investigated the sole association of each factor on *G. lamblia* infection risk. Multivariable models were subsequently built using variables with $P<0.1$ in age-adjusted models and the smallest associated quasi-likelihood value under the independence model criterion (QIC) for GEEs [38,39] on a complete data sample. The QIC criterion is an adaptation of the AIC (Akaike's information criterion) criterion for GEEs for model choice [40]. The final most parsimonious model using a complete data sample was fit back to the original data on as many observations as possible. Because longitudinal cohorts are subject to attrition at follow-up, we analyzed patterns in missing data for any *G. lamblia* infection and did sensitivity analyses. GEE estimations were based on the missing completely at random assumption [41].Random effects, based on maximum likelihood estimation, were also fit under missing at random assumption [38,41] and did not produce very different estimates in terms of ORs and standard errors [38,39]. Predictions for age-dependent risk of *G. lamblia* infections were displayed against raw data. Random-effects multinomial logit models were used to explore risk factors associated with assemblage, using a 4-categorical longitudinal outcome indicating no infection, assemblages A, B and mixed. Associations with risk factors were measured using relative risk ratios for each category against baseline and post-estimation relative risk ratios comparing B vs A, mixed vs. A, and mixed vs. B were then derived. The relative risk (RR) has a similar interpretation to that of OR. Predictions for the overall age-dependent probabilities of each assemblage were derived and displayed graphically. We used longitudinal panel data analysis [38] to explore the longitudinal effects of *G. lamblia* infections on risk of diarrhea and parameters of childhood growth (using z-scores for WAZ, HAZ, and BAZ) during childhood. These analyses were tailored to the nature of the outcomes, namely continuous longitudinal for growth parameter z-scores and a binary longitudinal outcome for the presence of diarrhea. In case of longitudinal outcomes, the GEE and mixed modelling estimation yielded similar findings, while for diarrhea risk, subject specific estimates are shown. Statistical significance was inferred by $P<0.05$. All statistical analyses were done using Stata version 17 (StataCorp, College Station, TX, 2021).

## Results

### Sample analysis

We analyzed 2,812 stool samples collected periodically from 498 (98.8% of the 504 children recruited– 6 children provided no stool sample) between 1 month and 8 years of age (1 [102 samples], 3 [256], 7 [351], 13 [347], 18 [137], 24 [317], 30 [195], 36 [376], 60 [356], and 96 [375] months). Median stool samples analyzed per child was 6 (Q1-Q3, 5–8). Among 953 *G. lamblia* positive samples by qPCR, assemblages A and B were identified using the novel SNP Taqman assay in 419 samples while 214 samples were genotyped using 18S rRNA sequence analysis. No other *G. lamblia* assemblage was detected among samples analyzed by 18S rRNA sequence analysis. A total of 320 samples could not be genotyped using either method.

### Age-dependent risk of infection with *G. lamblia* and with specific assemblages

*G. lamblia* infection was detected at least once in 79.7% of the 498 children during the first 8 years of life, and 30.7% tested positive at least once for assemblage A, 50.4% tested at least once for assemblage B and 6.4% had at least one mixed (both A and B) infection. Age-dependent risk of *G. lamblia* infections predicted by the longitudinal models against observed proportions are shown in Fig 1. The proportions infected increased non-linearly with a rapid increase during the first year of life followed by a more gradual increase reaching a peak at 60 months (raw data 53.9% infected vs. predicted 52.7% [95% CI 47.4–57.6%]) (Fig 1 and S2 Table). *G. lamblia* assemblage analysis showed that B was the dominant assemblage present in this sample. Age-dependent proportions infected with either of or both assemblages are shown in Fig 2. A small proportion of infections were with both assemblages. Infections peaked at 5 years for

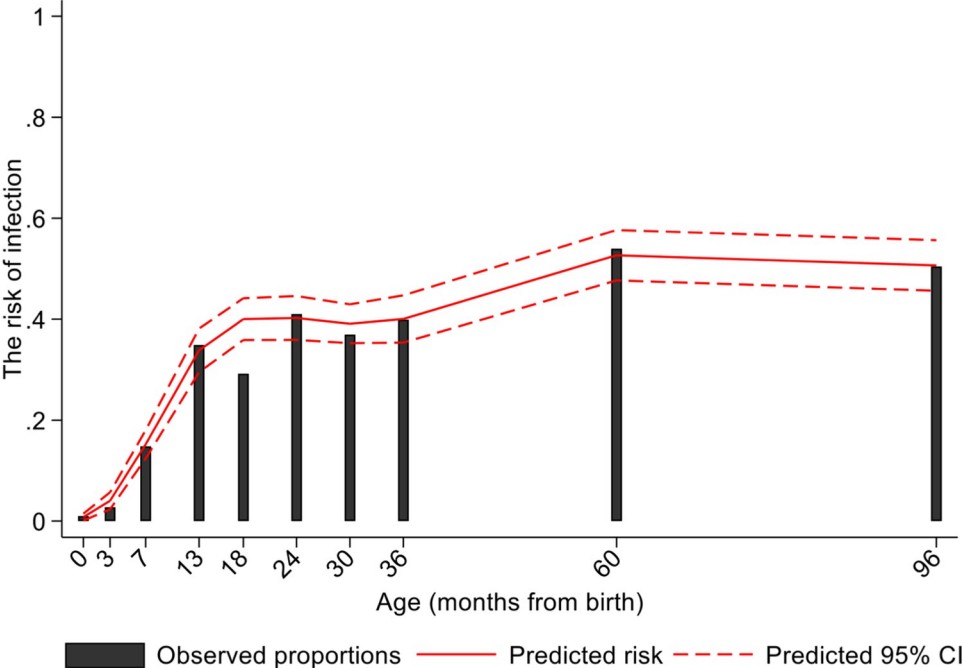

**Fig 1. Age-dependent predicted risk of infection with *G. lamblia* against the raw data.** A total of 2,812 stool samples were collected between 1 month and 8 years of age in a sample of 498 children from a birth cohort and analyzed for the presence of *G. lamblia* DNA by PCR. CI–confidence interval.

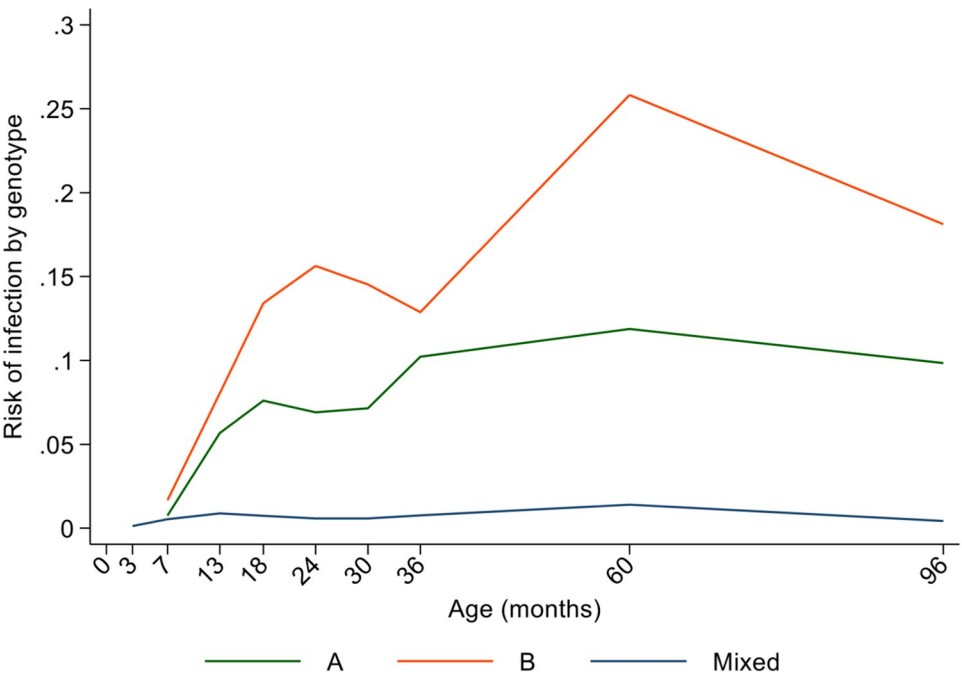

**Fig 2. Age-dependent risk of infection with *G. lamblia* stratified by assemblages A, B, or both (mixed).**

assemblage B and reached a plateau at 5 years for assemblage A. An analysis of 172 children with at least 2 *G. lamblia* infections during follow-up for which *G. lamblia* assemblage/s could be characterized showed 52.3% to have only infections with assemblage B identified, 18.6% with assemblage A only identified, 3.5% with mixed infections only, 18.6% with alternating infections with A and B, 4.1% with alternating infections between B and mixed (A and B), 1.2% with alternating infections between A and mixed, and 1.7% with alternating infections with A, B, and mixed (S2 Fig).

## Factors associated with age-dependent risk of *G. lamblia* infection

Distributions of individual, parental, and household characteristics for the 498 children and age-adjusted and multivariable associations of these with *G. lamblia* infections in childhood are shown in Table 1. In multivariable analyses, factors significantly associated with greater risk of *G. lamblia* infection were male sex (female vs. male, adj. OR 0.70, 95% CI 0.57–0.87, P = 0.001), daycare (adj. OR 1.98, 95% CI 1.50–2.63, P<0.001), childhood STH infections (adj. OR 1.71, 95% CI 1.35–2.17, P<0.001), maternal STH infections (adj. OR 1.27, 95% CI 1.03–1.58, P = 0.029), not having a household WC (WC vs. latrine, adj. OR 0.80, 95% CI 0.64–0.99, P = 0.040), and exposures to peri-domiciliary donkeys (age interaction, P = 0.034). Factors for which significant multivariable associations were observed with longitudinal risk of *G. lamblia* infection are illustrated graphically in S3 Fig.

## Factors associated with age-dependent risk of infection with *G. lamblia* assemblages

We explored the effects of childhood, parental, and household factors on the relative risk of infections with *G. lamblia* assemblages using a longitudinal outcome comparing A, B, and mixed infections (i.e B vs. A; mixed vs. A; mixed vs. B) among those children infected with *G.*

**Table 1. Age-adjusted and multivariable associations between childhood, parental, and household factors and risk of *G. lamblia* infection.**

| Variables | Category | N (%) | Age-adjusted | | | Multivariable | | |
|---|---|---|---|---|---|---|---|---|
| | | | OR | 95% CI | P value | OR | 95% CI | P value |
| **Childhood** | | | | | | | | |
| Sex | Male | 239(48%) | 1 | | | 1 | | |
| | Female | 259(52%) | **0.75** | **0.61–0.92** | **0.006** | **0.70** | **0.57–0.87** | **0.001** |
| Breastfeeding | 0–6 months | 59(11.9%) | 1 | | | | | |
| | 7–12 months | 179(35.9%) | 1.28 | 0.90–1.82 | 0.162 | | | |
| | >12 months | 243(48.8%) | 1.02 | 0.73–1.43 | 0.909 | | | |
| Excusive breastfeeding | Mean/SD months | 3.23/2.41 | 0.96 | 0.92–1.01 | 0.082 | | | |
| Birth order | 1st-2nd | 250(50.2%) | 1 | | | | | |
| | 3rd-4th | 168(33.7%) | 1.15 | 0.91 | 0.229 | | | |
| | > = 5th | 80(16.1%) | 1.28 | 0.95 | 0.111 | | | |
| Any daycare to 3 yrs | No | 408(81.9%) | 1 | | | 1 | | |
| | Yes | 90(18.1%) | **1.76** | **1.34–2.31** | **<0.001** | **1.98** | **1.50–2.63** | **<0.001** |
| **Anthelmintics (tv)** | No | 27(5.4%) | 1 | | | | | |
| | Yes | 471(94.6%) | 0.865 | 0.220 | 0.687 | | | |
| Any STH (tv) | No | 288(57.8%) | 1 | | | 1 | | |
| | Yes | 210(42.2%) | **1.78** | **1.41–2.23** | **<0.001** | **1.71** | **1.35–2.17** | **<0.001** |
| *A. lumbricoides* (tv) | No | 332(66.7%) | 1 | | | | | |
| | Yes | 166(33.3%) | **1.80** | **1.39–2.31** | **<0.001** | | | |
| *T. trichiura* (tv) | No | 395(79.3%) | 1 | | | | | |
| | Yes | 103(20.7%) | **1.60** | **1.18–2.17** | **0.003** | | | |
| **Maternal** | | | | | | | | |
| Age (yrs) | < = 20 | 142(28.5%) | 1 | | | | | |
| | 21–29 | 235(47.2%) | 1.00 | 0.78–1.18 | 0.987 | | | |
| | > = 30 | 121(24.3%) | 0.84 | 0.63–1.12 | 0.222 | | | |
| Ethnicity | Afro-Ecuadorian | 150(30.1%) | 1 | | | | | |
| | Non-Afro-Ecuadorian | 348(69.9%) | **0.72** | **0.58–0.91** | **0.005** | | | |
| Education | Illiterate | 67(13.5%) | 1 | | | | | |
| | Primary completed | 305(61.2%) | 1.08 | 0.78–1.49 | 0.638 | | | |
| | Secondary completed | 126(25.3%) | 0.78 | 0.55–1.12 | 0.179 | | | |
| Maternal STH | No | 270(54.2%) | 1 | | | 1 | | |
| | Yes | 222(44.6%) | **1.44** | **1.17–1.78** | **0.001** | **1.27** | **1.03–1.58** | **0.029** |
| **Household** | | | | | | | | |
| Area of residence | Urban | 354(71.1%) | 1 | | | | | |
| | Rural | 144(28.9%) | 0.85 | 0.68–1.08 | 0.177 | | | |
| Socio-economic status | Low | 133(26.7%) | 1 | | | | | |
| | Medium | 180(36.1%) | 0.99 | 0.76–1.29 | 0.923 | | | |
| | High | 185(37.1%) | 0.80 | 0.61–1.04 | 0.099 | | | |
| Overcrowding | <3 persons | 297(59.6%) | 1 | | | | | |
| | > = 3 persons | 201(40.4%) | **1.33** | **1.08–1.64** | **0.008** | | | |
| Agriculture (tv) | No | 249(50%) | 1 | | | | | |
| | Yes | 249(50%) | 0.86 | 0.70–1.06 | 0.144 | | | |
| Bathroom (tv) | Latrine | 168(33.7%) | 1 | | | 1 | | |
| | WC | 330(66.3%) | **0.80** | **0.65–0.98** | **0.030** | **0.80** | **0.64–0.99** | **0.040** |
| Monthly income (US$) | <1 basic salary | 374(75.1%) | 1 | | | | | |
| | >1 basic salary | 30(6.0%) | 0.75 | 0.47–1.19 | 0.222 | | | |
| House construction | Wood/bamboo | 102(20.5%) | 1 | | | | | |

*(Continued)*

**Table 1.** (Continued)

| Variables | Category | N (%) | Age-adjusted | | | Multivariable | | |
|---|---|---|---|---|---|---|---|---|
| | | | OR | 95% CI | P value | OR | 95% CI | P value |
| | Cement/brick | 393(78.9%) | 1.13 | 0.88–1.46 | 0.346 | | | |
| Material goods | 1–2 | 221(44.4%) | 1 | | | | | |
| | 3–4 | 277(55.6%) | 0.98 | 0.71–1.08 | 0.209 | | | |
| STH in household members | No | 326(65.5%) | 1 | 0.71–1.15 | 0.398 | | | |
| | Yes | 172(34.5%) | 0.90 | | | | | |
| **Peri-domiciliary animals** | | | | | | | | |
| Cats (tv) | No | 413(82.9%) | 1 | | | | | |
| | Yes | 85(17.1%) | 0.92 | 0.67–1.27 | 0.601 | | | |
| Dogs (tv) | No | 428(85.9%) | 1 | | | | | |
| | Yes | 70 (14.1%) | 1.05 | 0.73–1.49 | 0.809 | | | |
| Pigs (tv) | No | 257(51.6%) | 1 | | | | | |
| | Yes | 241(48.4%) | 0.94 | 0.76–1.18 | 0.605 | | | |
| Chickens (tv) | No | 55(11.0%) | 1 | | | | | |
| | Yes | 443(89.0%) | 0.95 | 0.80–1.13 | 0.590 | | | |
| Cows (tv) | No | 429(86.1%) | 1 | | | | | |
| | Yes | 69(13.9%) | 1.16 | 0.77–1.74 | 0.489 | | | |
| Horses (tv) | No | 407(81.7%) | 1 | | | | | |
| | Yes | 91(18.3%) | **2.22** | **1.12–4.41** | **0.022** | | | |
| | Age interaction | | **0.99** | **0.97–1.00** | **0.021** | | | |
| Donkey (tv) | No | 454(91.2%) | 1 | | | 1 | | |
| | Yes | 44(8.8%) | **3.23** | **1.25–8.37** | **0.016** | **2.99** | **1.11–8.07** | **0.030** |
| | Age interaction | | **0.98** | **0.96–1.00** | **0.016** | **0.98** | **0.96–1.00** | **0.034** |
| Mules (tv) | No | 431(86.5%) | 1 | | | | | |
| | Yes | 67(13.5%) | 0.75 | 0.50–1.14 | 0.179 | | | |
| Any equine (tv) | No | 385(77.3%) | 1 | | | | | |
| | Yes | 113(22.7%) | 1.59 | 0.87–2.89 | 0.132 | | | |
| | Age interaction | | **0.99** | **0.98–1.00** | **0.044** | | | |

Estimates show population-averaged estimates using generalized estimating equations. Age-adjusted analyses include polynomial terms of age up to the power of 5 (all P< = 0.02). Missing: breastfeeding (n = 17), maternal STH (6), monthly income (94), and household construction (3). Time varying (tv) variables are summarized as Yes vs. No during follow-up although estimates account for their variability over time. Data on household factors were collected around the time of birth of the child unless specified as tv. Anthelmintics–recent treatment with anthelmintic drugs (i.e. during previous 12 months). Overcrowding–persons/sleeping room. Material goods–number of household electrical goods. Pigs, chickens, cows, and equines–keeping these animals around house. Agriculture–child lives on a farm or visits a farm at least once a week. STH–soil-transmitted helminth infection. yrs–years. OR–Odds ratio. CI–confidence interval. Statistically significant findings (P<0.05) are shown in bold.

*lamblia* in whom an assemblage was identified. Findings of age-adjusted and multivariable analyses are shown in S3 Table. The primary effects of interest were those of potential risk factors on the risk of *G. lamblia* assemblage B vs. A. Table 2 shows age-adjusted and multivariable effects of childhood, parental, and household factors on the relative risk of infections with assemblages B vs. A. The presence of STH in the mother (OR 1.89, 95% CI 1.22–2.93, P = 0.005) and during childhood (any STH, OR 3.02, 95% CI 1.76–5.12, P<0.001; *A. lumbricoides*, OR 2.83, 95% CI 1.52–5.25, P = 0.001; *T. trichiura*, OR 2.57, 95% CI 1.23–5.38, P = 0.012) in the child were the only factors significantly associated with relative risk of having assemblage B versus A in age-adjusted and also in multivariable analyses (maternal STH, adj. OR 1.60l, 95% CI 1.02–2.53, P = 0.042; childhood STH, adj. OR 2.82, 95% CI 1.62–4.89, P<0.001).

**Table 2. Age-adjusted and multivariable effects of childhood, parental, and household factors on the relative risk (RR) of infections with *G. lamblia* assemblages B vs. A.**

| Variables | Comparisons | Age-adjusted | | | Multivariable | | |
|---|---|---|---|---|---|---|---|
| | | RR | 95% CI | P value | RR | 95% CI | P value |
| **Childhood** | | | | | | | |
| Sex | Female vs. Male | 0.99 | 0.64–1.54 | 0.961 | | | |
| Breastfeeding | 7–12 vs. 0–6 m | 1.35 | 0.65–2.80 | 0.420 | | | |
| | >12 vs. 0–6 m | 1.33 | 0.65–2.72 | 0.437 | | | |
| Exclusive breastfeeding | 1 month effect | 1.01 | 0.92–1.11 | 0.825 | | | |
| Birth order | $3^{rd}$-$4^{th}$ vs. $1^{st}$-$2^{nd}$ | 1.35 | 0.65–2.81 | 0.419 | | | |
| | $> = 5^{th}$ vs. $1^{st}$-$2^{nd}$ | 1.33 | 0.65–2.72 | 0.437 | | | |
| Any daycare to 3 yrs | Yes vs. No | 1.24 | 0.73–2.11 | 0.426 | | | |
| Anthelmintics (tv) | Yes vs. No | 0.89 | 0.52–1.55 | 0.690 | | | |
| *A. lumbricoides* (tv) | Yes vs. No | **2.83** | **1.52–5.25** | **0.001** | | | |
| *T. trichiura* (tv) | Yes vs. No | **2.57** | **1.23–5.38** | **0.012** | | | |
| Any STH (tv) | Yes vs. No | **3.02** | **1.76–5.12** | **0.000** | **2.82** | **1.62–4.89** | **<0.001** |
| **Maternal** | | | | | | | |
| Age (yrs) | 21–29 vs. < = 20 | 1.02 | 0.60–1.73 | 0.942 | | | |
| | > = 30 vs. < = 20 | 0.80 | 0.44–1.47 | 0.478 | | | |
| Ethnicity | Non-Afro. vs. Afro. | **0.51** | **0.32–0.82** | **0.006** | 0.68 | 0.38–1.23 | 0.205 |
| Education | Primary vs. Illit | 0.58 | 0.28–1.19 | 0.137 | | | |
| | Second vs. Illit | 0.45 | 0.20–1.01 | 0.052 | | | |
| Maternal STH | Yes vs. No | **1.89** | **1.22–2.93** | **0.005** | **1.60** | **1.02–2.53** | **0.042** |
| **Household** | | | | | | | |
| Area of residence | Rural vs. Urban | 0.77 | 0.47–1.27 | 0.308 | | | |
| Socio-economic status | Medium vs. Low | 0.81 | 0.46–1.42 | 0.458 | | | |
| | High vs. Low | 0.82 | 0.46–1.46 | 0.504 | | | |
| Overcrowding | > = 3 vs. <3 pers. | 1.24 | 0.79–1.93 | 0.353 | | | |
| Agriculture (tv) | Yes vs. No | 0.88 | 0.57–1.36 | 0.554 | | | |
| Bathroom (tv) | Yes vs. No | 0.89 | 0.55–1.45 | 0.632 | | | |
| Monthly income (US$) | >1 vs. <1 salary | 0.87 | 0.71–1.06 | 0.152 | | | |
| House construction | Cement vs. wood | 0.79 | 0.45–1.36 | 0.386 | | | |
| Material goods | 3–4 vs. 1–2 | 0.96 | 0.63–1.54 | 0.946 | | | |
| STH in household members | Yes vs. No | 0.68 | 0.40–1.14 | 0.143 | | | |
| **Peridomiciliary animals** | | | | | | | |
| Cats (tv) | Yes vs. No | 0.80 | 0.39–1.65 | 0.543 | | | |
| Dogs (tv) | Yes vs. No | 1.10 | 0.49–2.45 | 0.821 | | | |
| Pigs (tv) | Yes vs. No | 0.90 | 0.54–1.51 | 0.685 | | | |
| Chickens (tv) | Yes vs. No | 1.01 | 0.68–1.52 | 0.947 | | | |
| Cows (tv) | Yes vs. No | 0.99 | 0.38–2.58 | 0.984 | | | |
| Horses (tv) | Yes vs. No | 0.70 | 0.33–1.47 | 0.343 | | | |
| Donkeys (tv) | Yes vs. No | 0.50 | 0.16–1.54 | 0.226 | | | |
| Mules (tv) | Yes vs. No | 1.04 | 0.42–2.57 | 0.925 | | | |
| Any equine (tv) | Yes vs. No | 0.75 | 0.38–1.50 | 0.421 | | | |

Age-adjusted analyses include polynomial terms for age up to power of 5 (all P<0.001). Time varying (tv) variables. Data on household factors were collected around the time of birth of the child unless specified as tv. Material goods–number of household electrical goods. Pigs, chickens, cows, and equines–keeping these animals around house. Agriculture–child lives on a farm or visits a farm at least once a week. CI–confidence interval. STH–soil-transmitted helminth infection. Non-Afro–non-Afro-Ecuadorian. yrs–years. m–months. SES -socioeconomic status. Overcrowding–persons/sleeping room. Statistically significant findings (P<0.05) are shown in bold.

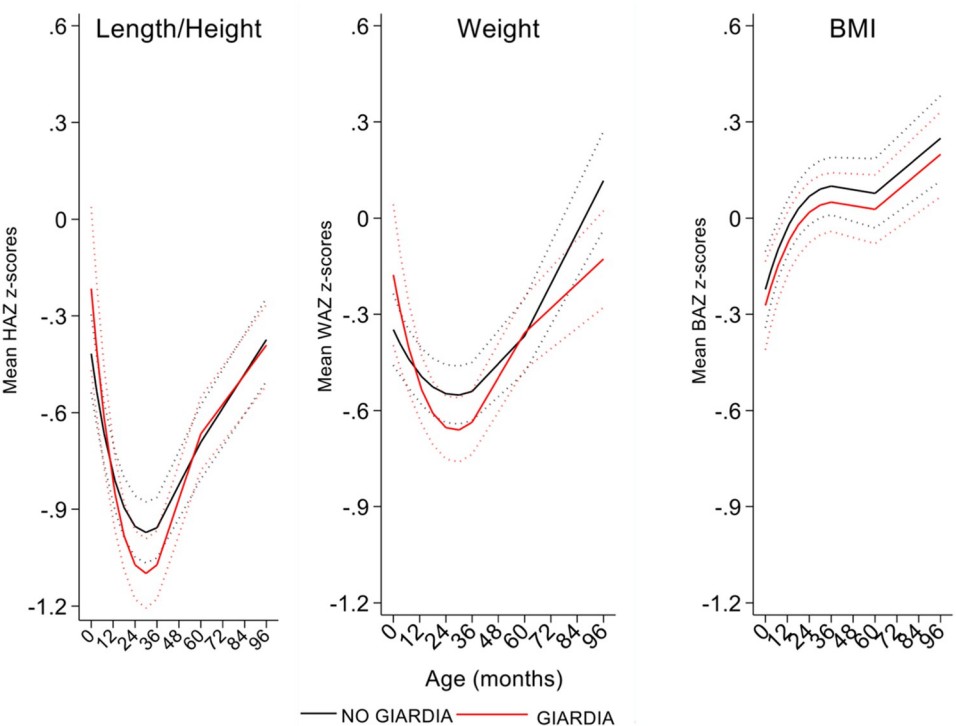

**Fig 3. Effects of *G. lamblia* infection on trajectories of height-for-age, weight-for-age, and BMI-for -age z scores during childhood up to 8 years.** Y axes show z scores for each of the growth parameters. BMI–body mass index. Interrupted curves represent 95% confidence intervals.

## Effects of infections of *G. lamblia* and assemblages on childhood growth trajectories growth and risk of diarrhea

Infections with *G. lamblia* during childhood had significant non-linear effects on growth trajectories estimated using weight-for-age (WAZ), height-for-age (HAZ), and BMI-for-age (BAZ) z scores (Fig 3). Estimates are provided in Table 3. Age trajectories for HAZ and WAZ showed significant interactions with age (HAZ, P = 0.025; WAZ, P = 0.017) and age$^2$ (HAZ, P = 0.025, WAZ, P = 0.011). Age-trajectory curves for HAZ were below WHO standards indicated the presence of stunting during the first 8 years of life in this sub-sample of the cohort. *G. lamblia* infections pulled both HAZ and WAZ score trajectories significantly below those representing the absence of *G. lamblia* during early childhood up to 5 years of age, after which differences were less relevant. Z-scores for the BMI curve associated with the *G. lamblia* infections remained below that of the absence of infection throughout childhood, although the effects were not significant. Sensitivity analyses using models for data collected up to 3 and 5 years matched subsequent trends for data up to 8 years, but with some gain in efficiency, namely smaller standard errors (S4 Fig). *G. lamblia* infections were associated also with an increased risk of diarrhea (RR 1.03, 95% CI 1.01–1.06, P = 0.011) (Table 3), particularly up to 3 years of age, after which the effect seemed to disappear (Fig 4). There was evidence of an age-interaction with *G. lamblia* on the risk of diarrhea (P = 0.06). If rather than as an outcome, diarrhea was analyzed as a time-varying exposure, diarrhea was associated with decreased WAZ (per episode, -0.092 z, 95% CI -0.181 - -0.003, P = 0.043) and BAZ (per episode, -0.136 z, 95% CI -0.253 - -0.018, P = 0.024) but not HAZ (per episode, -0.005 z, 95% CI -0.105 - -0.100, P = 0.926). These effects were independent of the presence of *G. lamblia*, although addition of

**Table 3. Effects of *G. lamblia* infections, age, and *G. lamblia*-age interactions (where appropriate) on probability of diarrhea and on growth trajectories for height-for-age, weight-for-age, and body mass index (BMI)-for-age z scores.**

| Variable | Height-for-age | | | | Weight-for-age | | | | BMI-for-age | | | | Risk of diarrhea | | | |
|---|---|---|---|---|---|---|---|---|---|---|---|---|---|---|---|---|
| | Estim | p-value | 95% CI | | Estim | p-value | 95% CI | | Estim | p-val | 95% CI | | RR/OR | p-value | 95% CI | |
| Age | -0.041 | <0.001 | -0.051 | -0.031 | -0.0152 | <0.001 | -0.0237 | -0.0067 | 0.0208 | <0.001 | 0.0113 | 0.0302 | 2.061 | 0.013 | 1.168 | 3.636 |
| *G. lamblia* (Y vs. N) | 0.201 | 0.127 | -0.057 | 0.460 | 0.1704 | 0.130 | -0.0505 | 0.3913 | -0.0499 | 0.195 | -0.1255 | 0.0256 | **1.033** | **0.011** | **1.007** | **1.058** |
| Age × *G. lamblia* | -0.026 | **0.025** | -0.048 | -0.003 | **-0.0232** | **0.017** | **-0.0423** | **-0.0041** | | | | | 0.983 | 0.058 | 0.965 | 1.001 |
| Age² | 0.0009 | <0.001 | 0.0006 | 0.0012 | 0.0003 | 0.007 | 0.0001 | 0.0005 | -0.0004 | <0.001 | -0.0007 | -0.0002 | 0.9994 | <0.001 | 0.9991 | 0.9997 |
| Age² × *G. lamblia* | 0.0006 | 0.0250 | 0.0001 | 0.0011 | 0.0006 | 0.011 | 0.0001 | 0.0010 | | | | | | | | |
| Age³ | -4.91e-06 | <0.001 | -6.8e-06 | -3.1e-06 | -1.09e-06 | 0.182 | -2.68e-06 | 5.09e-07 | 2.7e-06 | <0.001 | 1.1e-06 | 4.3e-06 | | | | |
| Age³ × *G. lamblia* | -3.78e-06 | 0.032 | -7.2e-06 | -3.3e-07 | **-4.04e-06** | **0.008** | **-7.00e-06** | **-1.08e-06** | | | | | | | | |

Estimates were derived from mixed models fit to continuous outcomes (height-for-age, weight-for-age, BMI-for-age) and from mixed binary models fit on diarrhea as a longitudinal binary outcome with *G.lamblia* presence as a time-varying covariate. The latter has a subject-specific interpretation as in the average effect of *G.lamblia* presence on risk of diarrhea. Statistically significant findings (P<0.05) are shown in bold. Age² –age to power of 2. Age³ –age to power of 3. RR–relative risk; OR–Odds ratio; CI–confidence interval; Y–yes; N–no; X–interactions.

*G. lamblia* to the model diminished the effects. Our data do not allow us to disentangle effects on growth of diarrhea due to presence of *G. lamblia* from effect of diarrhea attributable to other enteric infections. There was no evidence to suggest that different *G. lamblia* assemblages had any effect on growth trajectories or diarrhea risk (S4 Table).

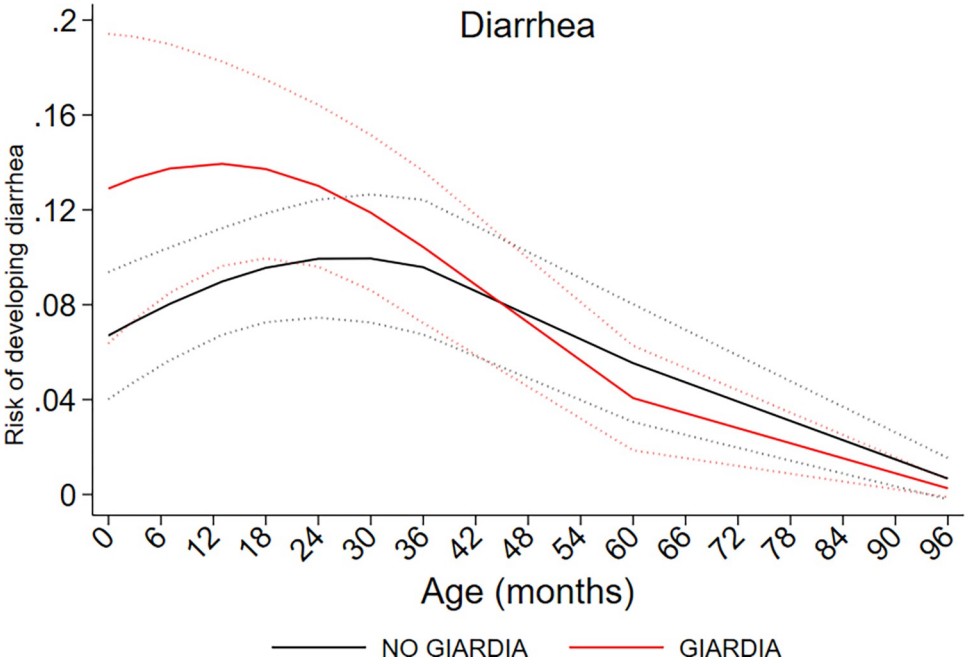

**Fig 4. The effects of *G. lamblia* infection on the probability of having diarrhea during early childhood up to 8 years.** Interrupted curves represent 95% confidence intervals. Interrupted curves represent 95% confidence intervals.

## Discussion

In the present analysis, we used a birth cohort from a rural district of tropical Ecuador to study longitudinally to 8 years the epidemiology of *G. lamblia* infection and A and B assemblages, the risk factors associated with infection, and the effects of infection on risk of diarrhea and growth trajectories. Our data show that *G. lamblia* was endemic in this setting with approximately 80% of children having at least one documented infection and proportions infected at any point in time exceeded 35% from 2 years of age with assemblage B being the dominant assemblage present. *G. lamblia* infections were associated with risk of diarrhea during the first 3 years of life and with transient impairment of weight and height trajectories between 1 and 4 years of age.

To our knowledge, this is the first longitudinal study to use more sensitive molecular methods to study the epidemiology of infection with *G. lamblia* and its assemblages in a Latin American setting. A previous birth cohort studying early childhood giardiasis was done at two sites in Latin America as part of the MAL-ED study, a multi-country birth cohort that included also participants from 6 sites in Africa and Asia [14]. The MAL-ED cohort enrolled more than 200 children per site at birth and followed them for 2 years with frequent collection of stool samples analyzed using an enzyme immunoassay to detect *G. lamblia* infections [14] or by quantitative PCR [2,23]. The study observed high rates of infection with a cumulative incidence that varied 37.7% to 96.4% between sites over the 2-year observation period [14].

Previous studies done in Latin America have shown a variable dominance of A versus B assemblages between studies [15,16,42], with occasional reports of human infections with other assemblages such as D [43,44], G [43] and C [45]. Previous molecular analyses of giardiasis done in Ecuador have examined the presence of *G. lamblia* assemblages and sub-assemblages [5,45,46], showing varying relative distributions of *G. lamblia* assemblages. A study done in a rural population of children living in the geographically contiguous district of Eloy Alfaro (in Esmeraldas Province) showed a dominance of assemblage B (61% were B, 32% were A, and 7% mixed) [5], while a study of schoolchildren living in highland and midland Andean and sub-tropical coastal communities showed a greater relative proportion of assemblage A [46]. Factors identified in this study to indicate a greater relative risk of infection with assemblage B versus A was the acquisition of STH infections during childhood and the presence of maternal infections around the time of birth of the child, which might reflect shared route/s of transmission or unknown host or environmental factors favoring assemblage B cyst dispersal [47] or survival in the environment.

We studied the potential effects of a variety of peri-domestic animals (including dogs, cats, chickens, cows, pigs, and equines) as potential sources of zoonotic *G. lamblia* infections in humans. Our time-varying analysis for peri-domiciliary animals allowed us to account for changes in exposures to these animals over time. The only animals associated with risk of childhood infection in this setting were equines with the strongest effect observed for donkeys and no relative effect by assemblage. Assemblages (and sub-assemblages) of A and B of *G. lamblia* have been shown to infect a wide variety of non-human hosts including horses [48] and donkeys [49]. Donkeys may be source of human *G. lamblia* infections in this study setting and may occur through close contacts of children with donkeys or contamination of the peri-domestic environment with donkey feces.

*G. lamblia* infections have been associated with persistent diarrhea in young children and asymptomatic infections in older children and adults living in endemic areas [18], while in industrialized settings, infections cause acute diarrhea at all ages among those exposed to infection during waterborne outbreaks [17] or through travel to endemic areas [50]. The association with acute diarrhea in endemic settings in low and middle-income countries (LMICs) is

unclear: a meta-analysis of observational studies of diarrhea in children living in endemic areas in LMICs indicated an overall protective effect of infection against acute diarrhea, except in very young children, but a 3-fold increased risk of persistent diarrhea [18]. Acquired but partial immunity against *G. lamblia* is considered to develop in humans, likely mediated by Th17 inflammatory mechanisms that include the induction of mucosal secretory IgA and other protective molecules [51,52]. Effects on diarrhea risk are likely to depend on age of first infection and intensity of reinfections: time to induction of immunity may vary by these factors with protection becoming established earlier in more highly endemic settings. Protection, however, is likely to be partial and primarily mediated against disease rather than colonisation and cyst carriage. In this study, using a highly sensitive molecular assay, we observed an increased risk of diarrhea associated with infection up to 3 years of age that might indicate the slow acquisition of protection against disease. Because of the timing of repeated stool collections, we were unable to distinguish persistent from reinfections. Diarrhea was all-cause so we cannot exclude interactions with other enteric pathogens such as rotavirus or enterotoxigenic *Escherichia coli* [53].The MAL-ED study, which did monthly stool collections during the first 2 years of life, did not observe an association with acute diarrhea, although persistent infections in the first 6 months of life were associated with reduced subsequent diarrheal rates at the study site with the highest incidence of infection (14). We were unable to detect effects of *G. lamblia* assemblage on diarrhea risk, a finding that is consistent with the published literature [54].

Giardiasis has been shown to be associated with impaired childhood growth in several cross-sectional and longitudinal studies [19,20]. Previous birth cohort studies in LMICs include: i) a study of 157 children followed up to 4 years in an urban slum in Fortaleza, Brazil, with routine stool sampling at 4-month intervals analyzed by microscopy, showed that symptomatic infections with *G. lamblia* were associated with lower WAZ and HAZ z scores [22]; ii) a cohort of 629 children followed to 2 years of age with monthly stool sampling in an urban slum in Dhaka, Bangladesh, showed that detection of *G. lamblia* (by immunoassay of stool) during the first 6 months of life was associated with decreased length-for-age (LAZ) z scores at 2 years [21]; and iii) the MAL-ED cohorts of 1,469 children followed up to 5 years with monthly stool sampling to 2 years and detection of *G. lamblia* using qPCR, showed that children with more frequent asymptomatic infections with *G. lamblia* or with higher parasite burdens, had significantly reduced LAZ scores at 2 years compared to children with infrequent infections or low parasite burdens [23]. No effect on LAZ scores was observed at 5 years [23]. These latter findings are consistent with our own in which we observed transient but significant effects of *G. lamblia* infections on reduced HAZ and WAZ z scores between 1 and 4 years. Interestingly, a recent analysis of the MAL-ED cohorts provided evidence that the effects of *G. lamblia* on linear growth during the first 2 years of life may be mediated through disruption of protein metabolism rather than through effects on intestinal permeability or mucosal inflammation [2]. The deficit in linear growth in under-5s appears to be transient—growth outcomes may approximate to those of less affected children from the source population in later childhood.

Strengths of the study was the longitudinal design with high rates of follow-up and repeated sampling of children to detect *G. lamblia* infections using a highly sensitive qPCR assay [29]. Our longitudinal approach allowed us to study the dynamics of *G. lamblia* infection during childhood and account for the temporal dependencies and hierarchical structure of the data while considering all complete observations in a unified manner thus minimizing loss of information. Our analytic approach allowed us to consider longitudinal infection status with *G. lamblia*, changes over time of various risk factors, and minimize risk of reverse causality by ensuring the directionality between infection and growth outcomes. Our study was unusual by

the long period of follow-up of 8 years allowing us to examine effects of infection on health outcomes into school age.

The study has several important limitations. Data on some risk factors were collected periodically to measure time-varying effects but did not include some variables potentially linked to longitudinal *G. lamblia* risk such as potable water and infections among household members. In the case of potable water, there were important changes in access to potable water in the study district over the study period that coincided with a period of economic growth and national investments in infrastructure including access to potable water [34]. We were unable to look at infection persistence because the periodicity of routine sample collection times did not allow us to distinguish persistent from reinfections. Further, lack of data on other enteric pathogens did not allow us to explore potential interactions with these on health outcomes or control for potential confounding by unmeasured pathogens. Our findings are likely to be generalizable to similar pediatric populations living in rural Districts in coastal Ecuador, but not necessarily to populations living in other regions of the country (e.g., highland Andean and lowland Amazon), where geographic, climatic, living conditions, and socio-cultural factors may be distinct. Although relatively large, the sub-sample of the birth cohort analyzed here had limited power for infrequent exposures and outcomes, and for sub-group analyses with relatively small samples such as relative effects of assemblages on health outcomes. Future analyses measuring effects on health outcomes of assemblage types or even sub-types would need to be done in large birth cohorts in highly endemic settings.

In conclusion, our data, from surveillance sample within a birth cohort, show a relatively high endemicity of infection in this population of children living in a rural tropical District in Ecuador, and evidence for early but temporary effects of infection on all-cause diarrhea risk, and transient impairment of height and weight during early childhood. Additional longitudinal studies are required in different geographic settings to study potential interactions of *G. lamblia* with other endemic enteric pathogens on diarrhea risk and long-term effects on linear growth.

## Supporting information

**S1 Fig. The 18S rRNA consensus sequence and SNPs (inside brackets) selected to genotype A versus B *G. lamblia* alleles using TaqMan probes.** N's correspond to SNPs and MNP's masked and not targeted.
(TIFF)

**S2 Fig. Assemblages A and B detected during follow-up in the 325 children with an identified assemblage.** Shown are proportions that during follow-up have assemblages A (A) or (B) alone detected, assemblages A and B alone detected at different times during follow-up (A-B), and those with mixed and non-mixed infections (Mixed).
(TIF)

**S3 Fig. Age-dependent predicted risk of infection with *G. lamblia* stratified by categories of factors showing statistically significant effects in multivariable analyses.**
(TIFF)

**S4 Fig. Effects of *G. lamblia* infection on trajectories of height-for-age, weight-for-age, and BMI-for-age z scores up to 3, 5, and 8 years.** Y axes show z scores for each of the growth parameters. BMI–body mass index. Interrupted curves represent 95% confidence intervals.
(TIFF)

**S1 Table. Sequences of primers and probes used in molecular analyses.**
(DOCX)

**S2 Table. Observed and predicted risk of G. lamblia infection by age.**
(DOCX)

**S3 Table. Age-adjusted and multivariable effects of childhood, parental, and household factors on the relative risk of infections with *G. lamblia* assemblages A vs. B and mixed (A and B) vs. A or B.** Age-adjusted analyses include polynomial terms for age up to power of 5 (all P<0.001). RR–relative risk. CI–confidence interval. Time varying (tv) variables. Data on household factors were collected around the time of birth of the child unless specified as tv. Material goods–number of household electrical goods. Pigs, chickens cows, and equines–keeping these animals around house. Agriculture–child lives on a farm or visits a farm at least once a week. STH–soil-transmitted helminth infection. Non-Afro–non-Afro-Ecuadorian. yrs–years. m–months. SES -socioeconomic status. Overcrowding–persons/sleeping room. Statistically significant findings (P<0.05) are shown in bold.
(DOCX)

**S4 Table. Effects of *G. lamblia* assemblage comparisons (among those with *G. lamblia* infection) on probability of diarrhea and on trajectories for height-for-age, weight-for-age, and body mass index (BMI)-for-age z scores.** RR–relative risk; OR–Odds ratio; CI–confidence interval; Y–yes; N–no; X–interactions. Statistically significant findings (P<0.05) are shown in bold.
(DOCX)

**S1 Data. Raw data used for analyses.**
(TXT)

## Acknowledgments

We thank the ECUAVIDA study team for their dedicated work and the cohort mothers and children for their enthusiastic participation. We acknowledge the support of the Ecuadorian Ministry of Public Health and the Directors and Staff of the Hospital "Padre Alberto Buffoni", Quinindé, Esmeraldas Province.

## Author Contributions

**Conceptualization:** Rojelio Mejia, Philip J. Cooper.

**Data curation:** Martha Chico.

**Formal analysis:** Tannya Sandoval-Ramírez, Irina Chis Ster.

**Funding acquisition:** Victor Seco-Hidalgo, Diana Garcia-Ramon, Philip J. Cooper.

**Investigation:** Tannya Sandoval-Ramírez, Victor Seco-Hidalgo, Evelyn Calderon-Espinosa, Diana Garcia-Ramon, Andrea Lopez, Manuel Calvopiña, Irene Guadalupe, Martha Chico, Rojelio Mejia, Irina Chis Ster, Philip J. Cooper.

**Methodology:** Tannya Sandoval-Ramírez, Victor Seco-Hidalgo, Evelyn Calderon-Espinosa, Diana Garcia-Ramon, Martha Chico, Rojelio Mejia, Irina Chis Ster, Philip J. Cooper.

**Project administration:** Evelyn Calderon-Espinosa, Andrea Lopez, Martha Chico, Philip J. Cooper.

**Resources:** Manuel Calvopiña, Rojelio Mejia, Philip J. Cooper.

**Supervision:** Victor Seco-Hidalgo, Martha Chico, Philip J. Cooper.

**Visualization:** Irina Chis Ster.

**Writing – original draft:** Philip J. Cooper.

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
