## [Decision Letter · Decision Letter 0]

31 Aug 2023

Dear Dr. Cooper,

Thank you very much for submitting your manuscript "Epidemiology of giardiasis and assemblages A and B and effects on diarrhea and growth trajectories during the first 8 years of life: analysis of a birth cohort in a rural district in tropical Ecuador" for consideration at PLOS Neglected Tropical Diseases. As with all papers reviewed by the journal, your manuscript was reviewed by members of the editorial board and by several independent reviewers. In light of the reviews (below this email), we would like to invite the resubmission of a significantly-revised version that takes into account the reviewers' comments. 

We cannot make any decision about publication until we have seen the revised manuscript and your response to the reviewers' comments. Your revised manuscript is also likely to be sent to reviewers for further evaluation.

Sincerely,

David Leitsch

Guest Editor

Abhay Satoskar

Section Editor

Reviewer's Responses to Questions

**Key Review Criteria Required for Acceptance?**

**Methods**

-Are the objectives of the study clearly articulated with a clear testable hypothesis stated?

-Is the study design appropriate to address the stated objectives?

-Is the population clearly described and appropriate for the hypothesis being tested?

-Is the sample size sufficient to ensure adequate power to address the hypothesis being tested?

-Were correct statistical analysis used to support conclusions?

-Are there concerns about ethical or regulatory requirements being met?

Reviewer #1: The study objectives are clear. See below regarding sampling interval and sample size.

Reviewer #2: PLOS Neglected Tropical Diseases

Epidemiology of giardiasis and assemblages A and B and effects on diarrhea and growth trajectories during the first 8 years of life: analysis of a birth cohort in a rural district in tropical Ecuador

--Manuscript Draft

The paper is written well technically and scientifically.

There three major issues commented below regarding the microscopic examinations, investigated loci and the assemblages, which they need to be revised.

The authors worthy commented some other limitation of the study, this should include also vegetables issues.

The paper contains useful informations and it will be an add to the public health significance of G. and G-sis.

Comments:

……Giardiasis, caused by the enteric protozoal parasite Giardia lamblia (also known as Giardia

86 duodenalis and Giardia intestinalis), is a neglected tropical disease with a worldwide

87 distribution [1] .

This comment is not valid any more, pl rephrase and update. Giardia is everywhere, it is not tropical, it is partially neglected, all of these neglected diseases are now top priority.

Also, you need to emphasize the role of drinking water and outbreaks transmission of G., update and take lit reviews in consideration. Same for vegetables contamination and transmission. (Introduction and discussion). 

101 endemic settings is less clear [17].Giardiasis is considered to cause stunting and/or failure-to-..

Pl keep space between after the dot.

142 temperatures generally ranging 23-32oC with yearly rainfall of around 2000-3000mm.

Keep space between units and words.

L 177-184, keep space between units and words.

Same in the following chapters

Question: what about microscopical analysis results? please, provide explanation for this

Question How about other assemblages. Please, provide explanation for this.

Usually, three loci are using to detect and characterize Giardia. This is a common agreement in the science of Giardia

Please, provide explanation for this

300 dependent risk of giardia infections predicted by the longitudinal models against observed

Pl correct species name

Linguistic issues and grammar corrections are needed

References

Pl revise, correct and unify the style of cited references. There are irregularities.

Reviewer #3: There are some details missing in the methodological description. 

specific points:

Page 10, line 169ff: I am not sure how this relates to Giardia or other protozoal diagnostics? Were there only STH detected or “all” parasites. Please include a statement for clarification.

Page 10, line 178: How much stool sample was processed? Was fresh stool or preserved stool or both processed?

Page 10, line 181/182: Please provide more information about the qPCR assay: Provide primer/probe sequences, please include statement why the assay was only using 7µl total volume (usually 25µl are used), was there a inhibition control PCR included (if not, why not?), check primer concentration (also in the other PCR assays)

Page 11, line 188ff: Please include primer and probe sequences of the newly developed assay. What are the parameters of the new assay (analytical sensitivity and specificity)? What was the positive/negative control DNA for the assay?

Page 11, line 197: what is “assay working stock”, please specify. 

Page 12, Line 219-221: Well, that is not as simple as Giardia represents a tetraploid parasite with assemblage B showing a high proportion of allelic sequence heterogeneity (also shows double peaks in chromograms). Please clarify.

I cannot judge whether the correct statistics were used.

**Results**

-Does the analysis presented match the analysis plan?

-Are the results clearly and completely presented?

-Are the figures (Tables, Images) of sufficient quality for clarity?

Reviewer #1: The results are overall well presented. However Table 3 lacks information in order to be readable on its own. For instance, age2 and age3 are not explained in the footnotes.

See also below.

Reviewer #2: Results are clearly presented.

some issues commented above

Reviewer #3: The results are overall well described. 

specific points:

Page 15, line 297-301: I found it difficult to follow the numbers. Please also include a supplementary table with all PCR data for each child, so that interested readers maybe able to use the data afterwards. Also make statement whether children stay Assemblage A or B positive or how many children show different assemblages in different longitudinal samples, etc. There is much more information in the dataset as now shown. Also interesting: Had the albendazole treatment some influence on the Giardia-positivity rates (Albendazole is an alternative Giardia treatment) .

Page 20, line 401-406: is this maybe an indication for zoonotic transmission of Ass B? Did the authors investigate the subtypes of assemblage AI and AII (AII being only found in humans, whereas AI might also be zoonotic)

Fig. 4: The Y axes should be risk of developing diarrhea, right? Please check.

**Conclusions**

-Are the conclusions supported by the data presented?

-Are the limitations of analysis clearly described?

-Do the authors discuss how these data can be helpful to advance our understanding of the topic under study?

-Is public health relevance addressed?

Reviewer #1: The discussion and conclusion is mostly well balanced. There could be more caution on small sub-groups and long time distance between sampling for some interpretations.

Reviewer #2: The complex epidemiological situation with Giardia offer space for myn discussions. However, the authors commented and discussed their results in accurate manner

Reviewer #3: PH is adressed and conclusions are not over-interpreted.

Discussion is sufficient and limitations are described.

specific points:

Page 24, line 485: “B” should probably mean “A”?

Maybe add a paragraph for assemblage B being the dominating assemblage type. What is the reason? 

How do the results relate to the Giardia data of the GEMS study (Kotloff, K.L., et al., The Lancet, 2013. 382(9888): p. 209-222.)?

**Editorial and Data Presentation Modifications?**

Reviewer #1: See below.

Reviewer #2: I suggested major revision because of some issues to provide an answer, the paper is well presented

Reviewer #3: Minor points

- There are some spelling errors that should be checked, eg.:

Page 4 line 56, there is an additional “in” before “coastal”; 

page 5, line 63, “which” should be probably removed; 

page 6, line 89 assemblies” should be “assemblages”;

**Summary and General Comments**

Reviewer #1: This is a longitudinal cohort of children in rural Ecuador. There is biannual follow-up until 3 years of age, and after this at 5 and 8 years. While the main research question was allergy STH infections, this sub-study investigates Giardia in stool samples of the children and the impact on child growth. The main finding is that presence of Giardia in stool is correlated to sanitary facilities at home, daycare attendance, male sex, and STH in both the child and mother. The presence of Giardia is shown to be a risk factor for impaired growth during the first three years of age. The study was well planned and conducted.

Some questions remain:

1. While the first sentence in Results in the abstract focuses on total number of stool samples, 2812, this obscures the fact that no time points had more than 376 delivered stool samples, and at 1 month only 102. This has implications for the interpretation of results. Comments on this is needed.

2. From the publication which the authors say describe the overall design and details of the study, altogether 2244 children had been followed up to 3 years. Still, the authors only included 504 children recruited in 2008 and 2009. I don’t see the rationale for this, since more participants would have strengthened the conclusions. Please explain. 

3. The authors include the presence of STH in their analysis, while other pathogens are not considered. This is particularly important when analysing the relationship between Giardia and diarrhea episodes. One further clarification is needed in this context; is the question of diarrhea only a question of present diarrhea at the time of sampling, or is it a question of diarrhea during a period of time. Since the authors have extracted DNA/RNA, have other pathogens been analysed? The presence of other common pathogens could potentially confound the role of Giardia. 

4. Assemblage comparisons: As far as I can understand, approximately 600 samples could be genotyped. Could seemingly lack of importance of either genotype be explained by the paucity of available genotyping at each time point?

5. Follow-up time points: Up to the age of 3 years the children are seen/give stool samples every 6 months. Even six months intervals are long intervals when investigating the role of Giardia in stunting, and interpretations should be cautious. But after the age of 3, the next sample comes two years later, and then 3 years after that again, at 8 years. While up to 3 years this study confirm previous trials, I think interpretations on the role of Giardia after this time is weak and could be omitted.

Reviewer #2: I provide my comments above

i would like to suggest the publication of the paper based on useful results and data after the major issues are commented

Reviewer #3: (No Response)
---

## [Decision Letter · Decision Letter 1]

7 Nov 2023

Dear Dr. Cooper,

We are pleased to inform you that your manuscript 'Epidemiology of giardiasis and assemblages A and B and effects on diarrhea and growth trajectories during the first 8 years of life: analysis of a birth cohort in a rural district in tropical Ecuador' has been provisionally accepted for publication in PLOS Neglected Tropical Diseases.

Best regards,

David Leitsch

Guest Editor

Abhay Satoskar

Section Editor

Reviewer's Responses to Questions

**Key Review Criteria Required for Acceptance?**

**Methods**

-Are the objectives of the study clearly articulated with a clear testable hypothesis stated?

-Is the study design appropriate to address the stated objectives?

-Is the population clearly described and appropriate for the hypothesis being tested?

-Is the sample size sufficient to ensure adequate power to address the hypothesis being tested?

-Were correct statistical analysis used to support conclusions?

-Are there concerns about ethical or regulatory requirements being met?

Reviewer #1: Yes to all- Issues regarding samples size and sub group analysis is addressed in the Dicussion.

Reviewer #2: reviewed previously

Reviewer #3: (No Response)

**Results**

-Does the analysis presented match the analysis plan?

-Are the results clearly and completely presented?

-Are the figures (Tables, Images) of sufficient quality for clarity?

Reviewer #1: Yes

Reviewer #2: reviewed previously

Reviewer #3: (No Response)

**Conclusions**

-Are the conclusions supported by the data presented?

-Are the limitations of analysis clearly described?

-Do the authors discuss how these data can be helpful to advance our understanding of the topic under study?

-Is public health relevance addressed?

Reviewer #1: Yes, this is improved.

Reviewer #2: reviewed previously

Reviewer #3: (No Response)

**Editorial and Data Presentation Modifications?**

Reviewer #1: (No Response)

Reviewer #2: (No Response)

Reviewer #3: (No Response)

**Summary and General Comments**

Reviewer #1: The revised manuscript with authors responses was at time difficult to follow since the line numbers didn't correspond to the responses. However the queries were addressed.

Reviewer #2: my comments have been answered

Reviewer #3: In the revision the authors adequately addressed all previous comments. I have no further questions.

PLOS authors have the option to publish the peer review history of their article (what does this mean?). If published, this will include your full peer review and any attached files.

Reviewer #1: No

Reviewer #2: **Yes: **Panagiotis Karanis

Reviewer #3: No

---

## [Editor Report · Acceptance letter]

15 Nov 2023

Dear Dr. Cooper,

We are delighted to inform you that your manuscript, "Epidemiology of giardiasis and assemblages A and B and effects on diarrhea and growth trajectories during the first 8 years of life: analysis of a birth cohort in a rural district in tropical Ecuador," has been formally accepted for publication in PLOS Neglected Tropical Diseases.

Best regards,

Shaden Kamhawi

co-Editor-in-Chief

Paul Brindley

co-Editor-in-Chief
